# Clustering of Long-Period Earthquakes Beneath Gorely Volcano (Kamchatka) during a Degassing Episode in 2013

**Sergei Abramenkov [1,2,\*], Nikolaï M. Shapiro [3,4], Ivan Koulakov [2,5,6,\*] and Ilyas Abkadyrov [2,6]**

[1] Institut de Physique du Globe de Paris, Université de Paris, CNRS UMR7154, 1 rue Jussieu, 75238 Paris, France

[2] Trofimuk Institute of Petroleum Geology and Geophysics SB RAS, Prospekt Koptyuga, 3, 630090 Novosibirsk, Russia; AbkadyrovIF@ipgg.sbras.ru

[3] Institut des Sciences de la Terre (ISTERRE), UMR CNRS 5375, Université Grenoble-Alpes, 38058 Grenoble, France; nikolai.shapiro@univ-grenoble-alpes.fr

[4] Schmidt Institute of Physics of the Earth, Russian Academy of Sciences, 119991 Moscow, Russia

[5] Laboratory of Seismic Imaging of the Earth, Novosibirsk State University, Pirogova 2, 630090 Novosibirsk, Russia

[6] Institute of Volcanology and Seismology FEB RAS, Piip Boulevard, 9, 693006 Petropavlovsk-Kamchatsky, Russia

\* Correspondence: abram.science@gmail.com (S.A.); koulakoviy@ipgg.sbras.ru (I.K.); Tel.: +7-913-4538-987 (I.K.)

**Abstract:** Gorely is one of the most active volcanoes in Kamchatka with a rich magmatic and eruptive history reflected in its composite structure. In 2013–2014, a temporary network of 20 seismic stations was installed on Gorely for one year. During the four months of its high degassing rate, seismic activity was mostly expressed in the form of a long-period (LP) seismic tremor. In this study, we have developed a workflow based on the combination of back-projection (BP), cluster analysis, and matched-filter (MF) methods. By applying it to continuous seismic records for the study period, we were able to identify discrete LP events within the tremor sequence automatically and individually investigate their properties. A catalog obtained using the BP detection algorithm consist of 1741 high-energy events. Cluster analysis revealed that the entire variety of LP earthquakes in this catalog could be grouped into five families, which are sequentially organized in time. Utilizing templates of these families in the MF search resulted in the complementary catalog of 80,615 low-energy events. The long-term occurrence of highly repetitive LP events in the same location may correspond to resonating conduits behaving in response to the high-pressure gases flowing from the decompressed magma chamber up to the volcano's crater.

**Keywords:** Gorely volcano; degassing; long-period seismicity; cluster analysis; back-projection; matched-filter

## 1. Introduction

Processes in active magma systems can set in motion different types of seismic sources occurring either through abrupt fractures of rocks, or oscillations of magma containing reservoirs, or as a combination of these two processes [1,2]. These sources generate seismic waves that can be recorded by seismic stations and used to monitor volcano activity and to diagnose the state of the magma plumbing system. Unlike purely tectonic earthquakes in non-volcanic areas, the volcano-related seismicity has a broad range of types starting from volcano-tectonic (VT) earthquakes with clear arrivals of the P

and S waves, to volcanic tremors, in which no distinct waves can be recognized [3]. If we factor the uniqueness of each volcanic region to such a specter of seismic signals produced even by a single volcano, one may see why the development of a unified classification is a genuine challenge for any seismologist.

This study is focused on long-period (LP) earthquakes. Among types of volcanic seismicity repeatedly described in the literature [2,4–6], these are troublemakers that make terminology confusing. Typically, LP events share a characteristic signature consisting of a brief high-frequency onset followed by decaying harmonic waveform that contains one or several dominant frequencies in the typical range of 0.5–5 Hz [7]. Such signal features are commonly interpreted as a broadband, time-localized pressure excitation mechanism (or trigger mechanism), followed by the response of a fluid-filled resonator [8]. In many cases, the LP volcanic earthquakes appear in swarms as a series of repetitive signals with almost identical waveform allowing to reconstruct the source geometry [9–16]. In Kamchatka, several clusters of deep and shallow repetitive LP events have been identified beneath the volcanoes of the Klyuchevskoy group that were activated synchronously with the occurrence of eruptions [17]. Studies of LP earthquakes precursory nature [18,19] have potentially immense importance for the public, especially in areas where volcanoes are located to close proximity of densely populated cities. An approach proposed in the present work allows us to detect individual LP events in continuous seismic records automatically. Furthermore, by using cluster analysis, we were able to reveal all possible variations of the LP seismicity occurred in the study region.

In the scope of this study, we investigate the LP earthquakes beneath the active Gorely Volcano in Kamchatka. Since 1984, the seismicity of Gorely is monitored by one permanent telemetered seismic station that was later supplemented with two other stations located on the neighboring Mutnovsky and Asacha volcanoes [20]. In 2013–2014, a temporary seismic network of 20 stations was installed on Gorely for one year. The analysis of data recorded by this network was used to obtain accurate locations of volcano-tectonic events beneath Gorely and to build a 3D seismic model [21]. A bright anomaly with a very high Vp/Vs ratio (up to 2) obtained in this study just below the summit of Gorely was interpreted as a shallow magma chamber. The upper limit of this anomaly at 2.6 km below the surface, followed beneath the topographic profile, might represent a level of the transition of fluids dissolved in the magma to gases due to decompression. This seismic velocity model also revealed a deeper anomaly of high Vp/Vs ratio located right below the shallow magma reservoir, which was interpreted as a conduit delivering the volatile-rich magma to the shallow reservoir from deeper sources. Both anomalies were surrounded by areas of low Vp/Vs ratio, with values reaching 1.4, which were interpreted as zones saturated with gases.

The primary purpose of this study is to further investigate the processes in the magmatic system beneath the Gorely volcano during the period of intense degassing activity in late 2013. In contrast to previously performed tomography study, here we use the continuous seismic records of the temporary network to study the distributions and properties of LP earthquakes beneath Gorely. Both the massive size of the dataset and the high expected occurrence-rate of these events urged us to develop the following three-step workflow: (1) identification of the most potent LP events by the back-projection detection technique, (2) cluster analysis of obtained catalog in order to group events with similar waveforms into several families, each represented by corresponding master event, (3) extension of the catalog to low-energy LP events by matched-filter detection technique using waveforms of the master events as a template. In the paper, we first give a concise overview of Gorely's geological context, followed by a brief description of available data. We describe each of the three steps mentioned above in the designated section and present the results of their implementation. Finally, we provide a possible interpretation of the resulting LP seismicity properties in terms of volcanic processes.

## 2. Gorely Volcano

Gorely is an active volcano located approximately 70 km away from Petropavlovsk-Kamchatsky, the most populated city on the Kamchatka Peninsula. Situated in the southern segment of Kamchatka's

Eastern Volcanic Front, 25 km from the Pacific coast, it is related to the ongoing subduction of the Pacific Plate, which is located at a depth of ~130 km below Gorely [22,23]. Morphologically Gorely is a compound shield-like stratovolcano with an altitude of ~1800 m above sea level and a relative elevation of ~850 m. Its upper part forms a linear northwest striking ridge of three merged primary cones and 11 superimposed summit craters complicated by more than 40 flank cones [24]. The modern Gorely edifice is located inside an ancient elliptic caldera with a size of 9 × 13 km, which is apparent on the topography map (Figure 1).

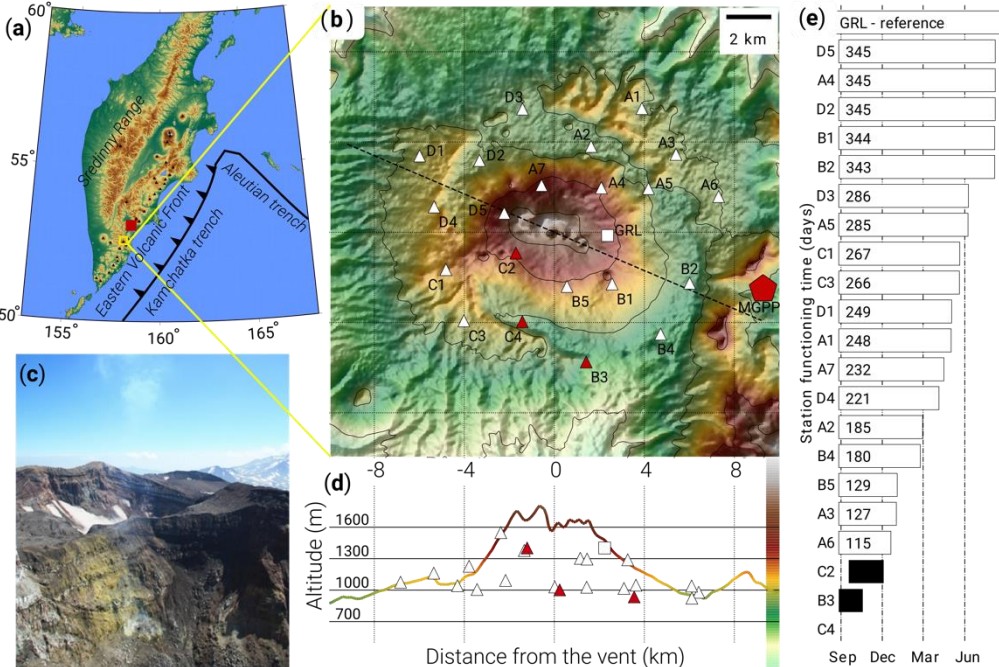

**Figure 1.** Gorely temporary seismic network (August 2013–August 2014) in the context of study region: (**a**) map of the Kamchatka peninsula with main tectonic features (red square indicates the city of Petropavlovsk-Kamchatsky, black triangles—active volcanoes and yellow square marks Gorely study region); (**b**) network geometry along with volcano topography (white triangles are stations used for analysis, white square—permanent GRL station, red triangles—nonfunctional stations, red pentagon—Mutnovsky Geothermal Power Plant); (**c**) photo of the volcanic vent taken during network installation process from the active crater edge; (**d**) projection of station locations on volcanic edifice along the dashed line; (**e**) data recovery chart sorted according to station functioning time.

The contemporary Gorely volcano represents the evolutionary development of an older volcanic center, followed by a radical transformation of its magma-feeding system [23]. Based on the age and composition of the erupted rocks, one can define three major stages of its formation [25,26]. The first (pre-caldera) stage is associated with the development of Pra-Gorely (also referred to as "Old Gorely"), which was an extensive (approximately 12 × 15 km in size) Middle-Pleistocene shield volcano stretched in the northeastern direction. Nowadays, the remnants of Pra-Gorely are mainly represented by peripheral parts of massive lava flows at the edges of the caldera and some relicts in the surrounding plateau [25]. The second stage led to the formation of a large caldera and massive felsic pyroclastic deposits in the surrounding area of 600 km$^2$. There is debate about whether this thick ignimbrite and pumice complex with the total volume >100 km$^3$ has been deposited during a single [25] or multiple caldera-forming eruptions ranging in age from 361 ka to 38 ka [27,28]. Regardless of the eruptions number, such depletion of a large magma chamber embedded in the Earth's crust below Pra-Gorely caused its roof to collapse. Limited by steeply dipping arc faults, the Gorely volcano caldera is a typical collapse structure of the Krakatau type that is confirmed by magmatic permeability of individual sections in the caldera boundary [25]. The last (post-caldera) major stage started toward the end of Late

Pleistocene with monogenetic volcanism on the weakened zone of the caldera rim. It was continued by the formation and development of the modern edifice in the central part of the caldera [26]. During Holocene Gorely's activity mainly consists of a cyclic alternation between phases Vulcanian-style explosive eruptions, voluminous (>0.1 km$^3$) lava flow eruptions, and intense degassing [25].

Having relatively high explosive eruption potential [29], Gorely may represent a significant hazard for aviation [30], tourists, and nearby infrastructures such as Mutnovsky Geothermal Power Plant (MGPP on Figure 1b) with a capacity of 50 MWt which provides a significant part of the electrical energy to the Petropavlovsk-Kamchatsky city and its surroundings. Therefore, the volcano has been thoroughly investigated by specialists in different disciplines of geosciences. The first robust data on the geological structure and development of Gorely volcano were published in [31,32], with the descriptions of the caldera, the associated pumice-ignimbrite deposits, the structural and material composition of the pre-caldera complex and the modern edifice. The comprehensive reconstruction of its Holocene activity via tephrochronological analysis was presented in [33]. In 1974–1977 a geological survey on a scale of 1:50,000 had resulted in a detailed geological map of the Gorely volcano [25]. In 20th century all eruptions (1921–1931; 1959–1960; 1980–1981; 1984–1986) were moderately explosive (VEI < 3) and occurred through the central summit with emission of basaltic-andesitic ash [24]. After the most recent one in 1986, a large fumarole was formed in the crater, through which an active emission of gases ensued. In the period of strongest degassing activity in 2010, the mass of gases emitted through this fumarole was estimated at 11,000 tons per day, with the outlet temperature reaching 900 °C. It is determined that these gases were composed of water (93.5%), $CO_2$ (2.6%), $SO_2$ (2.2%), HCl (1.1%), HF (0.3%), $H_2$ (0.2%), as well as some bromine and iodine compounds. It was estimated that under this regime, Gorely emitted about 0.3% and 1.6% of the total global fluxes from arc volcanism for $CO_2$ and HCl, respectively [34].

## 3. Seismic Data

In 1980, the Kamchatka Branch of the Geophysical Survey (KBGS) installed one telemetric seismic station GRL on the eastern slope of Gorely volcano. Two more permanent stations were installed in the summer of 2008 on the neighboring Mutnovsky and Asacha volcanoes. All stations were equipped with three-component sets of short-period channels based on SM-3 seismometers for recording the ground displacement velocity in the frequency band of 0.8–20 Hz. These permanent stations were used to investigate the seismicity beneath Gorely since 1984 [20,35]. Note, however, that these studies could only provide count and energy estimates for the events, but not the information about their locations.

A dense temporal seismic network (Figure 1) consisting of 20 three-component broadband seismographs was deployed on the Gorely volcanic edifice and its surroundings in August 2013 by joint efforts of scientists from Trofimuk Institute of Petroleum Geology and Geophysics SB RAS (initiated this project), Department of Geology and Geophysics of Novosibirsk State University (provided seismic instruments) and Institute of Volcanology and Seismology FEB RAS (provided logistical support of the fieldwork). The network was removed in August 2014 and provided ~350 Gb of continuous seismic records in total. Each of the temporal stations consisted of a CME-4311 (R-sensors, Moscow, Russia) three-component broadband sensor and a digital recorded Baikal-ACN-87/88 (R-sensors, Novosibirsk, Russia) with power supply provided by one box of 10 high-capacity power batteries Baken VTs-1 (UralElement, Verchniy Ufaley, Russia), external GPS antenna and necessary ventilated protection against dust and moisture. Baikal-ACN series recorders are three-channel autonomous seismic stations of an extended frequency range with an internal or external GPS module, a USB 2.0 channel for communication with a laptop, and a memory slot for SD card supporting volumes up to 32 GB. The CME-4311 three-component broadband velocimeter is built of three orthogonally oriented molecular-electronic transducer, and an electronic board, placed in a protective outer casing. The manufacturer stated flat instrument response in the frequency band of 0.016 (60 s) to 50 Hz.

We use the seismic records from a single permanent station GRL located on the volcanic edifice and maintained by the Kamchatka Branch of the Geophysical Survey as a reference. Even though

ground waters eventually flooded some stations of the temporal array, overall data coverage remained consistent enough for more than eight months with maximal spatial density during the four starting months. It allows us to analyze the seismicity on Gorely during a significant period containing an episode of volcano's intense degassing. In this study, we consider the period from the 28th August till the 17th December, when the maximum seismic volcano-related activity occurred beneath Gorely. During this period, 18 stations of the temporary network were functioning, providing dense observation system on the volcano.

Preliminary analysis revealed that the dominant part of seismic energy was emitted by numerous LP earthquakes, which occurred on average twice per minute during most active phases of the degassing episode. Corresponding signals have a duration of about 10–15 s long, with an energy peak at 3 Hz. The strong similarity of these waveforms for consecutive events is the most notable feature of the dataset. Figure 2a presents an example of a five-minute seismogram of vertical components recorded by all available stations that clearly shows the LP swarm beneath Gorely. In this interval, at least eight events can be visually identified, and all of them have almost identical waveforms, as seen in an example in Figure 2b,c.

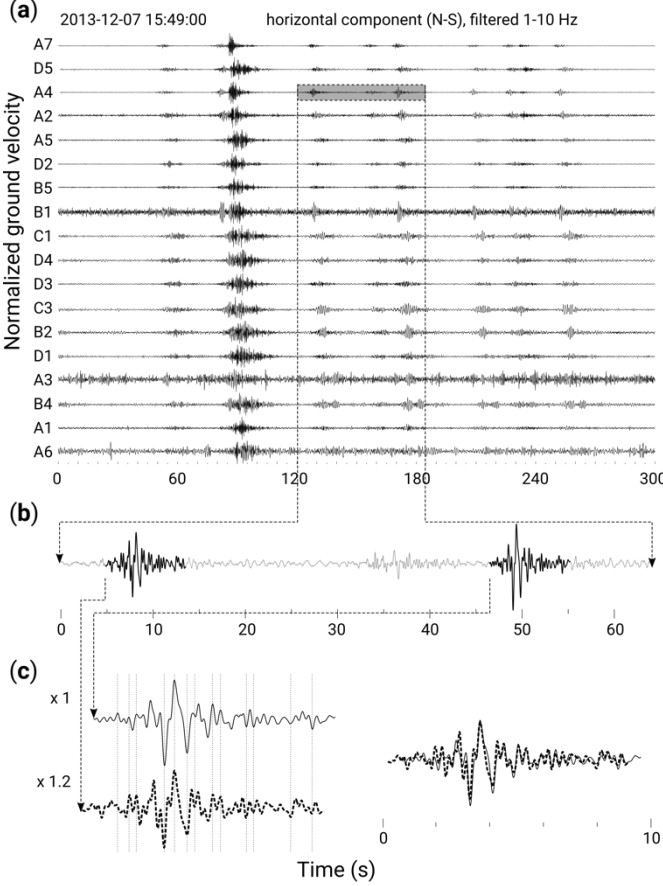

**Figure 2.** Example of LP swarm on Gorely: (**a**) 5-minute horizontal component records of 18 temporary stations; (**b**) zoom on representative LP signals at station A4; (**c**) close comparison of two waveforms.

A large number of events to be identified with a massive amount of the continuous seismic records urged us to develop a particular automated approach for compiling and analyzing the LP earthquakes. A robust catalog of specific type earthquakes is something highly desirable in seismology because it can give us valuable insights about the underlying mechanism and its evolution in time. We were trying to construct one by using three methods that successively built, investigate, verify, and enhance a catalog of the LP earthquakes. To build the first catalog with the most energetic events, we used

an automatic detection algorithm based on back-projection (BP) technique. We then investigated the acquired catalog via cluster analysis that gave us a set of templates for several LPs clusters and served as a verification tool. Finally, applying a matched-filter (MF) technique, we searched for less-energetic events with waveforms similar to the obtained templates, thus compiling the complementary extended catalog of low-energy LP events. This catalog, however, should be used as a supplementary one in the analysis since it could be noise-contaminated.

## 4. Back-Projection Detection and Location Algorithm

The back-projection (BP) method is a practical approach to detect and locate seismicity by taking advantages of a seismic network or array. The core idea of of this method is a stacking of seismic records shifted by precomputed travel times to the theoretical origin points followed by a grid-search for the local maximum in space and time. Several shift-and-stack methodologies have been described in literature starting from "semblance analysis" [36] and "source scanning algorithm" [37,38]. In this paper, we use "beamforming" version of the BP method suggested for studying tectonic low-frequency earthquakes, which share many signal features with long-period volcanic earthquakes [39].

A seismic event originated in the location $\vec{x}^*$ at the time moment $t^*$ is recorded by a set of receivers located in $\vec{r}_i$ ($i = 1, 2, \ldots$, N) as a set of waveforms $u_i(t)$. Assuming velocity model $v$ for a study region, we can compute theoretical travel-times $\tau(\vec{r}_i, \vec{x})$ between each receiver and some virtual source location $\vec{x}$. The BP technique, which general concept is schematically demonstrated in Figure 3, is based on the stacking of normalized signal envelopes shifted in accordance to these precomputed travel-times. Thus, the recorded wavefields are kinematically projected back to the point $\vec{x}$. We are using a bending algorithm from the Local Tomography Software (LOTOS) [40] for raytracing and calculating of travel times, which gives us the potential to improve results confidence by using a more realistic velocity model. In this study, our primary goal is the detection and only then relative location of LP seismicity, thus a simplified model with a constant velocity value equal to 2 km/s was used. For the case of Gorely, this appears to be suitable because the LP seismicity is generated at shallow depths within the volcanic edifice.

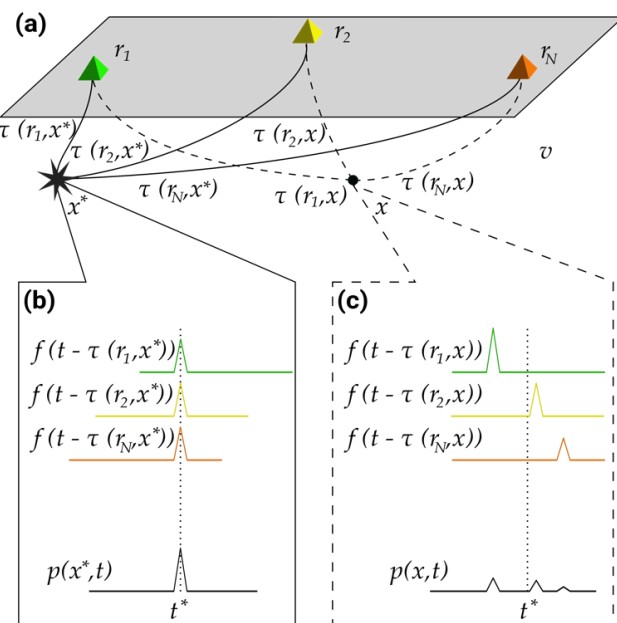

**Figure 3.** Back-projection general concept: (**a**) a simplified scheme of receivers and travel-times precomputed using ray-bending in the proper velocity model for two points of a study region; (**b**) back-projection to the actual location of the seismic event; (**c**) back-projection to the virtual source point with incorrect travel-times.

As described in the following paragraphs, for each point in the 3D space and time, we calculate a likelihood function that shows the coherency of the shifted waveforms across all receivers. Due to the small-scale media heterogeneity, the waveforms from an earthquake recorded at different stations are not coherent. That is why we ignore the signal phase and use a characteristic function (CF), $f_i(t)$, instead of raw records. Common examples of CF are high-order statistics of the seismic signal (Kurtosis, Skewness) [41], short-term average to long-term average (STA/LTA) ratio [42], and signal envelope [43]. After many trials with different types of CF, we found that in our case, the most optimal form is an energy envelope or an absolute values of seismogram smoothed in a moving window:

$$f_i(t) = \frac{1}{2h} \int_{-\tau_h}^{\tau_h} |u_i(t + \xi)| d\xi, \tag{1}$$

where $\tau_h = 3\,s$ is a half-size of the moving window for smoothing, and $u_i(t)$ is a horizontal component record of the $i$-th station.

We define back-projection intensity (BPI) function $p(\vec{x}, t)$, at point $\vec{x}$ as a stack of CFs $f_i(t)$ normalized for geometrical spreading $A(\vec{r}_i, \vec{x})$ and shifted according to travel-times between this point and each receiver:

$$p(\vec{x}, t) = \frac{1}{N} \sum_{i=1}^{N} A(\vec{r}_i, \vec{x}) f_i\left(t - \tau(\vec{r}_i, \vec{x})\right), \tag{2}$$

where $N$ is the number of used receivers. The geometrical spreading $A(\vec{r}_i, \vec{x})$ for a virtual source point $\vec{x}$ and a receiver located in $\vec{r}_i$ is calculated as follows:

$$A(\vec{r}_i, \vec{x}) = \frac{r_0}{d(\vec{r}_i, \vec{x})}, \tag{3}$$

where $r_0$ is the source size, which is approximated in our case by a unit sphere and $d(\vec{r}_i, \vec{x})$ is the length of the ray path in the reference model between the source and receiver.

For an actual source position (Figure 3b), the CFs calculated in Equation (2) are correctly shifted back in time and are stacked constructively, forming a maximum at the event's origin time. For any other points (Figure 3c), the same procedure will result in lower values of BPI. Computing BPI for a grid of virtual-source points $\vec{x}_j \in \vec{X}$ allows us to obtain an array of time-dependent functions $p(\vec{x}, t)$. Each of these functions represents a transformed wavefield kinematically projected back to a specific location inside the study region. They form a spatio-temporal distribution of BPI-$p(\vec{x}, t)$, which thereby may be considered as a time series of spatial images (snapshots) defined on the grid $\vec{X}$. Each snapshot $p(\vec{x}_j, t)$ depicts the likelihood of finding seismic source inside the study region at the specific time moment. In Figure 4, we present an example of a BPI snapshot corresponding to an average event based on the Gorely experimental data.

Incorporation of the BPI procedure inside a grid-search strategy is a core part of the BP detection technique. From full spatio-temporal BPI distribution $p(\vec{x}, t)$ we construct a compressed BPI:

$$p_c(t) = \max_{\vec{x}_j \in \vec{X}} \left[ p(\vec{x}_j, t) \right], \tag{4}$$

and use it as a detecting function. By scanning through $p_c(t)$ in time for local maxima larger than a threshold value $p_d$, we can effectively obtain time moments $t_{max}$ that correspond to the local maxima of the full BPI distribution. Then, we use a snapshot of BPI distribution at the $t_{max}$ moment for estimation of source location in 3-D space. Computing the time interval $\tau_w$ between the absolute maximum to the nearest local minimum after the detected event gives us an approximate value of the signal duration. In the case illustrated in Figure 4, resulting length of the event's signal was approximately 16 s. For earthquakes located outside the study region, the snapshot maxima are usually observed on

edges of the grid. We use this criterion in conjunction with the limitation on a signal duration $\tau_w$ to exclude teleseismic and slab-related earthquakes out of the catalog.

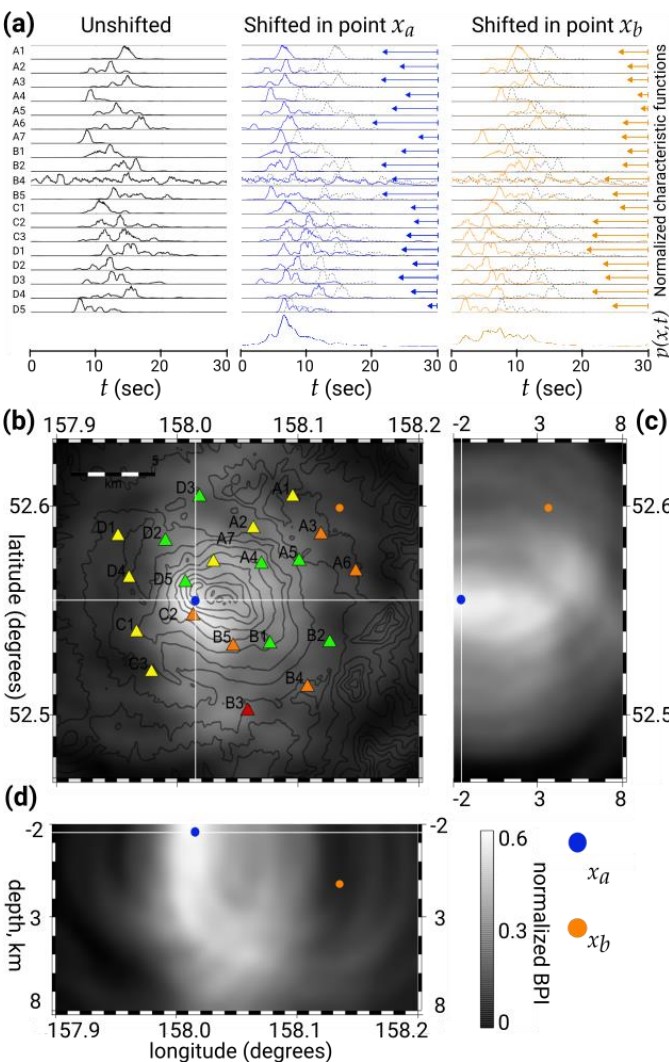

**Figure 4.** Example of the BP detection procedure for a high-energy LP event on Gorely: (**a**) signal envelopes (CFs) shifted according to precomputed travel-times; (**b**) horizontal, and (**c,d**) two vertical slices of 3-D BPI snapshot for a time of local maximum in detector function.

The BP detection technique was implemented to build a catalog of LP earthquakes beneath Gorely using two subsets of the whole data. Initially, with event detection threshold $p_d = 2$ µm/s, we performed detection for nine months of data available for five most consistent seismic stations (Figure 5a). This result showed that the significant of detections is condensed in the four starting months, with minor activity after middle December 2013. For this period of intense degassing, we were able to use data from 18 stations and obtain 9691 detections (Figure 5b). We then performed the BPI procedure with more conservative conditions presuming a more substantial value of the threshold equal to 4 µm/s, which led to decreasing the catalog size to only 1741 most energetic events (Figure 5c). Spatial distributions of the detected events (third column in Figure 5) show that the significant part of the detected LP seismicity is located right beneath the volcano edifice. These results point out similar characteristics of seismicity in time for both low and high energy parts of the catalog. It is also apparent that after December 7th, the activity on Gorely is rapidly decaying.

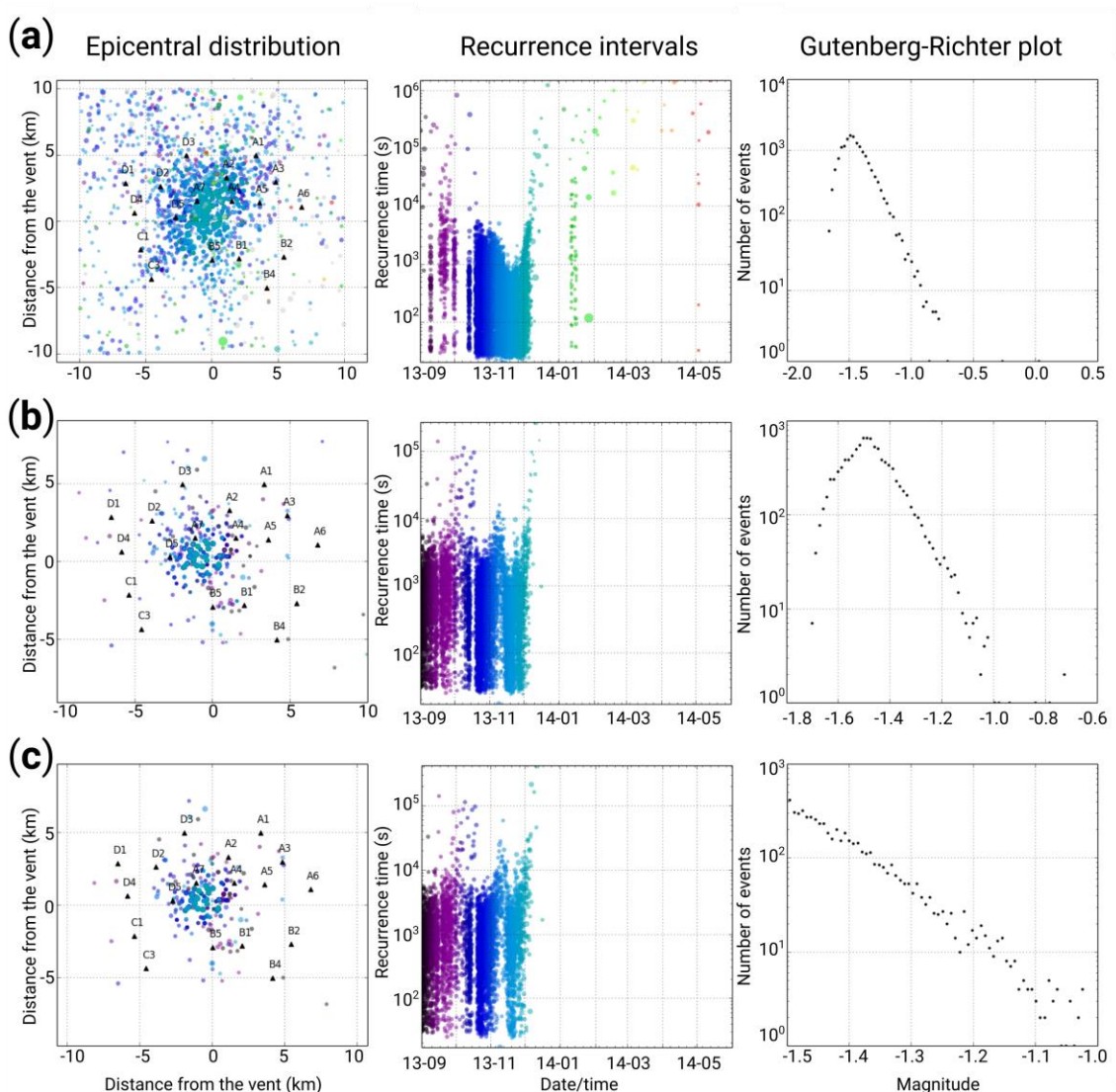

**Figure 5.** Results of applying the BP detection algorithm to Gorely data: (**a**) detections acquired using records of five temporary stations which functioned nine months; (**b**) same for 18 stations working during the four starting months, when degassing activity was exceptionally strong; (**c**) filtered detections obtained with 18 stations, which follow Gutenberg–Richter law and constitute BP-based catalog.

## 5. Cluster Analysis

The catalog of most-energetic events (Figure 6) derived from the implementation of the BP technique can be further investigated in detail via cluster analysis. By dividing earthquakes into groups, we can effectively reveal the overall seismicity structure that can be related to the characteristics of underlying processes. To do so, one needs a well-defined principle to group 'similar' earthquakes together. Our approach for similarity quantification between pairs of events, in general, resembles the one used for swarms of repeating long-period earthquakes at Shishaldin Volcano in Alaska [44].

For each pair of detected earthquakes, the corresponding waveforms of vertical components are cropped in a time window $\tau_w$ after the respective origin time ($\tau_w = 16$ s, in our case). We define the similarity of two earthquakes $l$ and $m$ as a correlation coefficient (CC) computed between the waveforms and mean averaged for all stations. To see the relationships between all events, one may plot these coefficients as a matrix, where each row or column reflects how similar the selected earthquake to the other ones in the catalog. Figure 7a shows the calculated CC matrix for the set of 1741 events identified for the Gorely volcano at the BP step with the higher threshold.

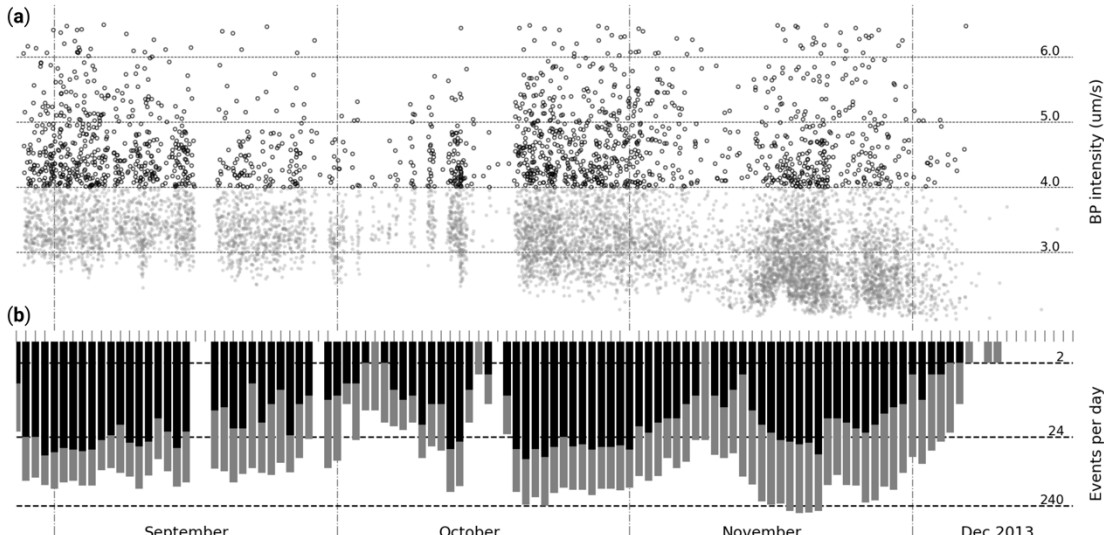

**Figure 6.** BP-based catalog of high-energy events (in black, correspond to Figure 4c) and indistinct detections (in grey, correspond to Figure 4b): (**a**) intensity and (**b**) daily rate of detection during the chosen period.

Despite the general similarity of all LP events in the BP-based catalog, some subgroups of events can be distinguished from the visual analysis of the CC matrix in Figure 7a. The diagonal of this matrix represents auto-correlations. We can see from four to six square patterns of high CCs along diagonal, that indicates a group of similar events localized in time. Some of the 'squares' are prolonged to off-diagonal part, suggesting the existence of subgroups inside. The number of groups of events with different properties can be approximately identified by estimating the rank of the CC matrix, which can be done by computing its eigenvalues. The resulted distribution of eigenvalues ranged in the decreasing order (Figure 7b) shows that only a small part of the first eigenvalues was large enough, while the rest is close to zero. We decided to take into account only five eigenvalues that are larger than 5% of the maximum one, thus estimating $N_c = 5$, the number of meaningful earthquake groups in the catalog.

Figure 7c presents the mean average CC of every detected LP earthquake with all other 1740 events in the catalog. The majority of them have the CC between 0.3 and 0.4 showing relatively high similarity of all events. At the same time, there are a few events having the correlation of around 0.1. We manually inspected all these events and identified that they are relatively short-duration VT earthquakes having completely different waveforms compared to the LP events. It does not mean that no other VT events occurred during the studied period because the used parameters of the BP method were specially adapted for searching the LP events (reference velocity, time window, frequency of filtering). Thus, after excluding these four VT events, a verified initial catalog of the LP earthquakes contained 1737 events with the highest energy.

To separate all detected LP earthquakes in five groups, we followed the iterative approach described in [45]. To find out a reasonable partition of initial clusters, we consequently excluded groups of similar earthquakes from the catalog. First, we calculate a mean average CC for every single earthquake with all other events across the whole catalog, as shown in Figure 7c. Next, an earthquake with the maximum average CC is taken as a master event for the first group. All events having the CC with the master event larger than a particular threshold $h$ are excluded from the catalog to form the first cluster. The procedure is repeated for the rest of the catalog: on each step earthquake with maximum average correlation is taken to form the next initial cluster that is excluded from the catalog until we get all $N_c$ clusters. After such selection, each master event is close to others in its group while staying far from the other master events. Depending on the chosen threshold, the entire catalog may be completely divided into $N_c$ clusters, or some earthquakes may stay ungrouped. In the second step, the defined

clusters iteratively resorted in accordance with the CC matrix. On each iteration, we first check every grouped event and place it in the group where it has the highest similarity to the corresponding master event. After that, the new master event for each cluster is determined by computing a new vector of the mean similarity from a subsection of the CC matrix. As a result, stable cluster distributions are organized after several iterations. Since this method converges to a local minimum, the final result depends on the cluster's starting 'centers' (master events). Reasonable choice of the starting cluster composition via excluding them from the catalog helps us form highly diverse clusters.

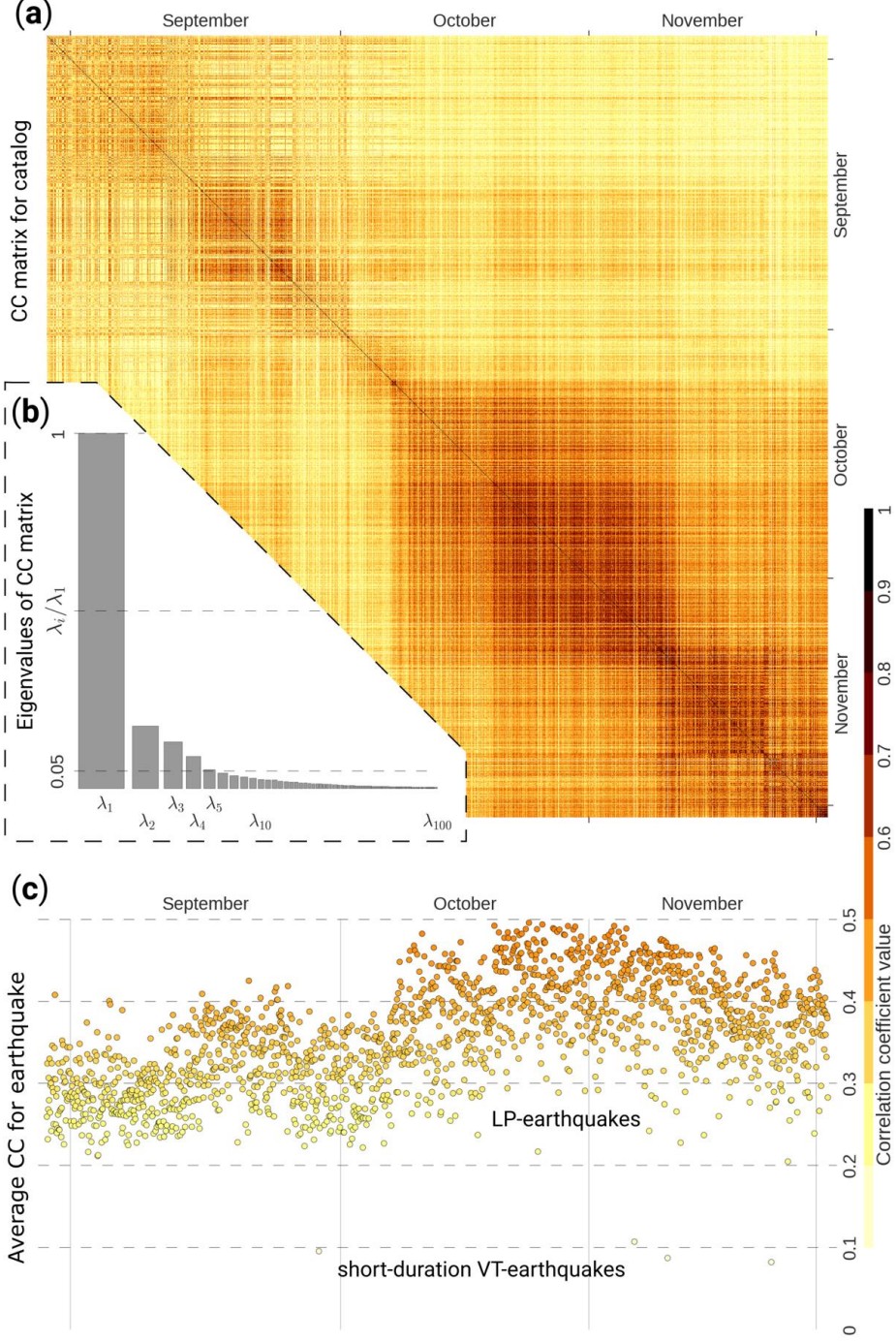

**Figure 7.** Similarity assessment of earthquakes from the BP based catalog: (**a**) waveform CC matrix; (**b**) first 100 eigenvalues of the matrix—sorted and normalized; (**c**) mean average similarity for each earthquake inside the catalog.

We performed described cluster analysis with the CC threshold of 0.3 and found that it converges to the stable distribution after the 3rd iteration. The CC values within the five groups and their time distributions are shown in Figure 8a,b. It can be seen that the CC values within distinct groups are higher than CCs with all events shown in Figure 7c, which demonstrate the adequacy of such classification of events. Final clusters are sequentially arranged in time that may indicate the possible evolution of seismic source properties or changes in the seismic velocity structure of the volcano.

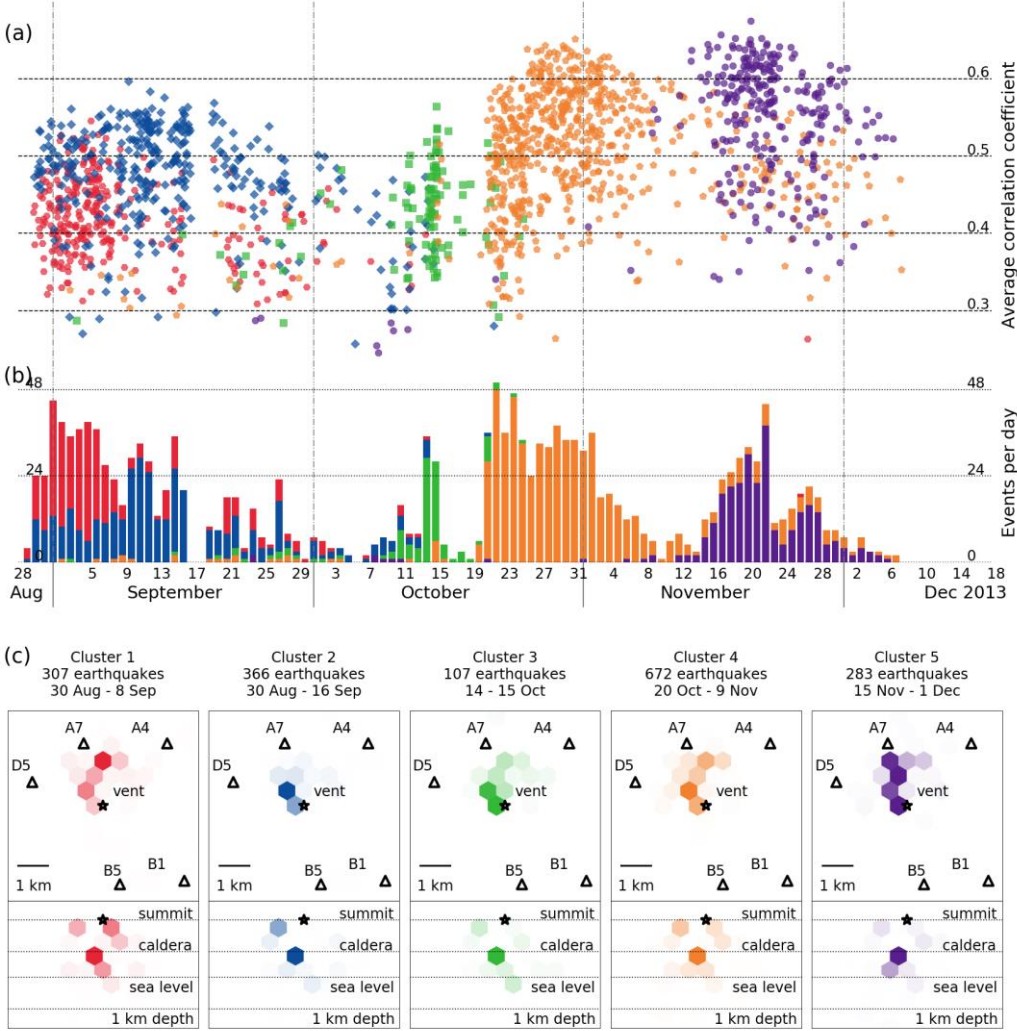

**Figure 8.** Cluster analysis results for BP based catalog. In all panels, the different colors indicate five identified clusters. (**a**) Average correlation coefficients for the final distribution after the 3rd iteration with five sequential clusters. (**b**) A daily number of events for different clusters. Note that each cluster has a dominant period with a maximum number of events per day. (**c**) Hexagonal plots of the events space distribution for each cluster projected to map view (upper row) and to vertical section oriented in west-east direction (lower row). Each hexagon presents confidence area of the location, while color intensity reflects normalized number of the events in this location.

## 6. Matched-Filter Detection Algorithm

The results of the cluster analysis allow us to create a set of templates that reflect common waveform features for all earthquakes in a certain group of the catalog. We create the cluster templates by stacking waveforms with the weights equal to their correlation coefficients. Thus, for the $i$-th receiver, the resulting template waveform $u_i^k(t)$ of the $k$-th cluster is computed as follow:

$$u_i^k(t) = \frac{\sum_{n=1}^{N_k} C_n^k u_i^n(t)}{\sum_{n=1}^{N_k} C_n^k}, \tag{5}$$

where $u_i^k(t)$ is a waveform of the *j*-th event inside the cluster, $C_n^k$ denotes event-to-master CC, and $N_k$ stands for the cluster size. Stacking increases signal-to-noise ratio so that we may treat the template as a fingerprint of a composite event with the common source mechanism for all events in the corresponding cluster. Since a template represents a generalized image of a cluster, we can compare it to another one visually and numerically by calculating CCs between them. In Figure 9, we show examples of the composite waveforms in some stations corresponding to the selected five groups of events.

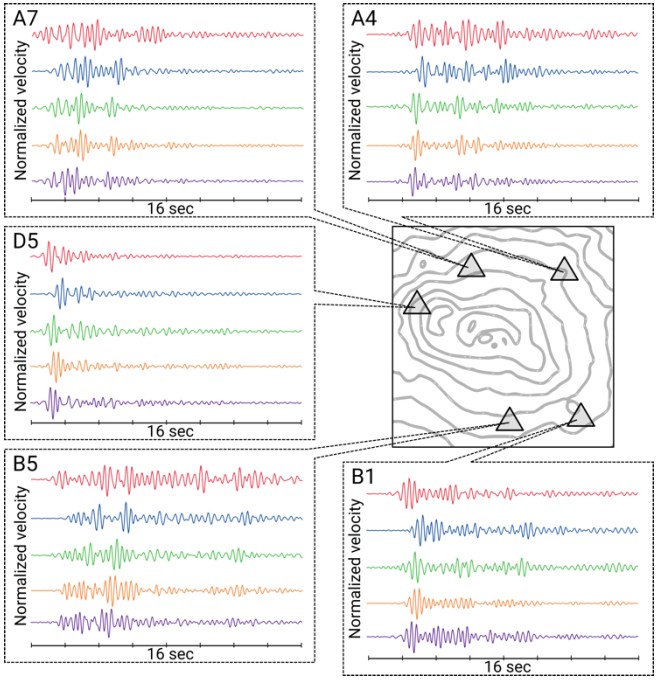

**Figure 9.** Template waveforms for five closest to summit stations (same colors as Figure 8).

We can use the constructed templates to search for other earthquakes, similar to ones selected for the corresponding clusters. The matched-filter (MF) search, which we use to detect multiplets [42], has become the standard way to identify families of tectonic low-frequency earthquakes [46–50] and volcanic LP earthquakes [17]. The MF algorithm consists of a matching template to continuous seismograms by computing CCs between the template and the waveforms in a sliding window. Comparing the derived time-dependent CC with a predefined threshold $c_d$, we can identify all events with the waveforms similar to the template, which creates a more extensive catalog than initially selected. Depending on the template-to-template similarity and chosen threshold, the same event can be detected with different templates. In these cases, we attribute such events to the template group having the highest CC. Unlike the BP technique, the MF approach is sensitive to the shape of seismic signals rather than its amplitude so that we can form a complementary catalog of weak LP-earthquakes. For the Gorely case, we were able to identify 80,615 LP earthquakes divided into five clusters. The time distributions of these events can be compared with high-energy events of the initial catalog in Figure 10.

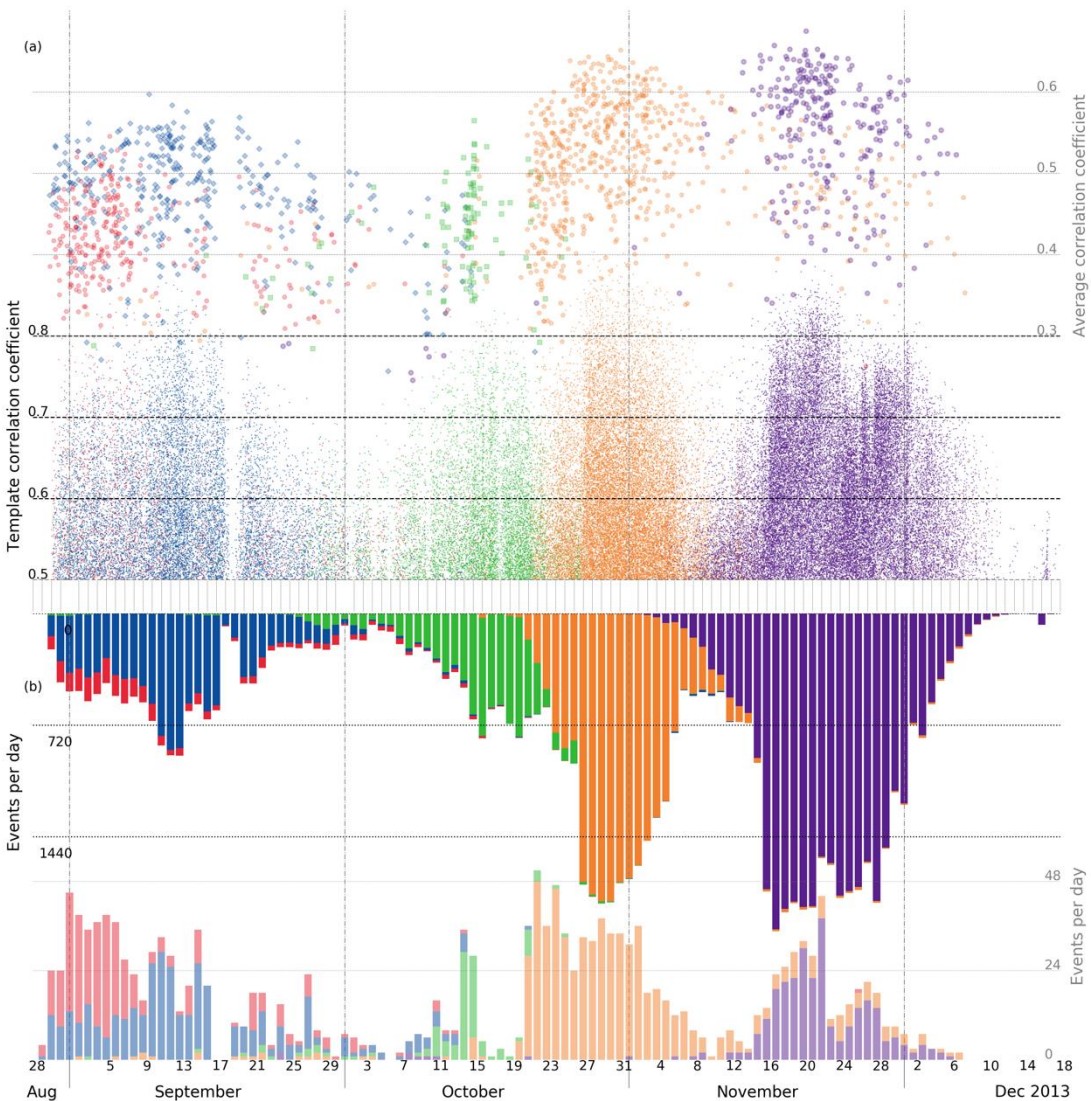

**Figure 10.** The MF-based catalog of lower-energy events in comparison with cluster analysis results for the BP-based catalog containing most-energetic earthquakes (in pale colors): (**a**) CCs for both catalogs (note that these are not the same type); (**b**) time distribution of the events' daily amount.

## 7. Discussion

Two implemented detection techniques gave us the possibility to carefully explore LP seismicity beneath Gorely volcano during a period of intense degassing. Our implementation of the BP approach is sensitive to the event energy, while the MF method is responsive to the waveform shape itself. By identifying signals of individual LP earthquakes in continuous seismic records of 18 temporary stations, we obtained the BP-based catalog containing 1737 high-energy events and the extensive MF-based list of 80,615 detections. It is important to note that the latter being complementary to the initial BP catalog is not entirely independent, as we are using templates constructed from the waveforms of identified events to obtain MF detections.

Cluster analysis of the BP-based catalog has demonstrated the limited variety and high repetitiveness of LP seismicity taking place beneath Gorely in the observation period. In total, we have identified five distinguishable families of LP earthquakes, which were sequentially arranged in time. In Figure 7, one can see the final distribution of high-energy LP earthquakes that gives us insight into the development of the conduit structure over time. At the beginning of the observation on August 28th, we see that the "red" family was dominating. By September 9th, its intensity has

incrementally decreased, as it was gradually replaced by the "blue" family, which in turn reached the maximum on September 10th and then weakened during a couple of weeks in the same manner. Between October 3rd and 7th, there is a gap in the LP seismic activity. Between October 9th and 16th, there was a short-lived peak of the "green" family activity followed by another gap on October 16–18. On October 20th, the "orange" family abruptly started and produced the largest group of the LP events, which gradually increased until November 9th. After another short gap, the last "violet" family started on November 13th, reached its peak on November 22nd, and then completely decayed to December 6th. After this moment, LP activity only occurred infrequently.

The extensive MF-based catalog provides additional information about the evolution of Gorely's degassing system. As one may see in Figure 10, lower-energy events are repeating the same pattern as their high-energy counterpart. However, there are fewer gaps in the activity, and the transitions between LP earthquake families are much smoother. Other differences are the domination of the "blue" family over the "red" one, considerably prolonged duration of the "green" period and the minor burst of detections on December 16th.

Each LP earthquake family produced a series of nearly similar signals emitted from a localized source region. These source regions were located beneath the volcano summit at depths of less than 1 km below the surface. In the recent tomography study of Gorely [21], the top of the prominent anomaly, representing the magma chamber, was located at ~2.5 km below the surface. The interface between very high Vp/Vs in the magma chamber and low Vp/Vs in the overlying carapace is interpreted as a level of the phase transition in the molten magma. We suppose that dissolved fluids were degassing at this stage due to lower pressure in the shallower part of the magma reservoir. The LP earthquakes identified in our study may indicate the following pathway of ascending high-pressure gases. The rapid degassing and fast dynamic propagation of gas bubbles through the conduit root may lead to self-sustained oscillations within the magmatic channel [51], generating the LP radiation recorded by seismic stations. It is possible that at some moment, the conduit structure changes, resulting in new oscillation parameters and characteristics of repeated LP events. Therefore, the observed evolution of LP earthquakes may reflect the structural changes in the shallow part of the volcano-magmatic system.

Alternatively, given generally shallow levels of LP radiation on Gorely, groundwaters may be involved in two-phase 'steam and water' resonator system in a similar way that was proposed for Ngauruhoe volcano in New Zealand [52,53]. The climate of Kamchatka with a heavy snow cover of volcanoes over a half-year and the glaciation of Pra-Gorely caldera both support this version. However, the chemical content of the and the rate of Gorely degassing implies that the proposed 'bubble-dynamic' mechanism of LP is primarily caused by dissolving magmatic fluids in the conduit root.

We see that only at the beginning of the observation period, two of LP earthquake families (the "red" and "blue" one) functioned at the same moments with about similar intensity of the high-energy events. In other periods, only a sole family is dominating at the time, which is apparent in the MF-based detection distribution (Figure 10). It probably means that the preferable degassing regime of Gorely requires only one conduit acting at a time. Smooth transitions in the number of detected events between families (Figure 10) and the structure of CC matrix (Figure 7a) imply that each dominant family gradually evolves into the next one. This may be interpreted as the slow migration of the source along a constricted pathway of magmatic gasses ascent. Close likelihood of template waveforms for master events of "blue", "green", "orange", and "purple" families also support this point, while "red" one represents alternative explanation. The co-existence of two LP families acting in the same period (but not simultaneously) may be interpreted as the balancing stage of the gas ascent process. In such conditions, the pressure gradient allows only a portion of gas bubbles to overcome constriction in the "blue" family origin point, while the rest of the gasses have to proceed laterally (Figure 8c) to the "red" family origin point.

**Author Contributions:** I.K. and N.S. designed the field experiment; I.A. and S.A. performed the field experiment; S.A. analyzed the data; N.M.S. and I.K. contributed to interpretation of the results. All authors participated in discussions of the results and contributed in writing the manuscript and preparing figures. All authors have read and agreed to the published version of the manuscript.

**Funding:** This study was supported by the Russian Ministry of Education and Science (grant #14.W03.31.0033), and by the European Research Council (ERC) under the European Union Horizon 2020 Research and Innovation Programme (grant agreement 787399-SEISMAZE). I.K. is supported by the Russian Foundation for Basic Research Grant #18-55-52003.

**Acknowledgments:** We are grateful to William Frank, Natalia Poiata, and Jean Soubestre for the valuable discussions about methods used in this study. We also thank the field experiment team members: Andrey Jakovlev, Pavel Kuznetsov, Evgeny Deev, Arseny Ivanov, and Alejandro Diaz-Moreno, who helped during installation of the temporary seismic network. We also appreciate the support of KBGS staff, who contributed to the operation of the permanent seismic network in Kamchatka. We thank Geneviève Moguilny for the attentive assistance during numerical computations which were partly performed on the S-CAPAD platform, IPGP, France. Finally, we acknowledge anonymous reviewers for their valuable comments that not only improved this manuscript but also pointed out some prospective directions for future work.

**Conflicts of Interest:** The authors declare no conflict of interest.

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
