# Peer review of "Clustering of Long-Period Earthquakes Beneath Gorely Volcano (Kamchatka) during a Degassing Episode in 2013"

_geosciences, doi:10.3390/geosciences10060230_

Round 1

Reviewer 1 Report

Review of geosciences_811652 by Abramenkov et al.

This is a very interesting paper using some excellent analysis and tools.  I do think the writing needs modest improvement which I have tried to outline where possible, but in general the paper is understandable, figures are mostly good and interpretation is consistent with the data.  I think the authors would also benefit from filling out the reference list a bit more to include a wider range of observations and context for LP earthquake families.  I would add the following:

Green and Neuberg,  2006 JVGR

Repeating earthquakes related to viscous andesitic magmas

Petersen et al, 2007, JVGR

Earthquake families discrimination

Hurst et al, 2014, JVGR

Repeating clockwork earthquakes before a phreatic eruption.

Battaglia et al, 2016 GRL

Earthquake families in a strombolian centre.

I also note that the cross matrix analysis would benefit from a bit of discussion around off diagonal higher correlation features in the 5 distinguished families.  By comparison, a very shallow swarm at Ngauruhoe volcano, New Zealand showed a distinct progression of earthquake frequency content and several distinct families.  The swarm showed a seasonal increase in event  productivity, possibly related to ingress of seasonal snowmelt (Jolly et al, 2012, JVGR),  Subsequent analysis (Park et al, 2019, JVGR)) found several event families and a distinct evolution pattern where a family evolved into another family through time.  Given the similarity of families found in the matrix off diagonal of the Gorely dataset.  I think it is important to evaluate if the events of that sequence may include a systematic evolutionary pattern beyond the productivity of one event to another.  You could test this in the same manner as outlined in Park et al. and with the advantage of more precise locations, you could constrain the source process.

Given the generally good condition of the manuscript and high quality of the observations, I would recommend something between minor and moderate revision including some thought around the points raised in this review.

Kind regards.

Additional comments:

20 ‘….with a rich….

27 ….A catalogue obtained….

29 ‘regrouped’ implies that there was a prior grouping or classification

40 ‘hybrid manner’….. this may confuse your audience as ‘hybrid’ earthquakes are specific to volcano-seismology for events sharing tectonic and LP features (e.g. Lahr, JVGR, 1994).

62 telemeteric < telemetered

69 …parallel to the topography line…, do you mean ‘…..followed beneath the topographic profile’?

Figure 1 this figure needs an inset showing the main cities, and some of the volcanic ark in the region of P-P-K.  Also the caption should mention the reason for different colours for stations.

115 Falling….do you mean ‘steeply dipping’?

124-26-  The general cultural features need to be represented in a proposed inset map in Figure 1.

133  XX < ‘the 20th century’?? and ‘moderate’ < ‘moderately’

136 ‘started to take place’ < ‘ensued’

138-139 CO2 < CO2 and other species corrected.

147  Are these sensors measuring velocity or displacement, what is the sampling rate, dynamic range etc.??

152- describe the sensor make model and characteristics sampling rate/effective pass-band etc.

186 ‘applying a Matched Filter (MF) technique’

186-188 What about noise contamination

189  Is the title correct here…this seems to be used both as a detection and location algorithm?

Figure 3 implies ray bending but the text implies straight line ray-paths.  Suggest reconciling the figure or the text.  Also is there any a priori information about the selected velocity model.

223 ‘as follows’

243 ‘resulting length’

281 and 311        ‘averaged’  might say mean or median depending on what is used?

Figure 7 and 311    It seems that some of the cluster ‘families’ seen in your cross-matrix are possibly related as sub-families as alluded to on line 311.  In other words, why don’t you interpret the off-diagonal parts of the matrix?

Figure 8c How do the spatial distributions reflect errors?....In other words, are the location errors smaller than the hexagons in all cases….This is a matter for clarity of your audience who may over interpret these plots.

374 ‘…seismicity occurred beneath….during a period of …..’

385 conduits < the conduit  (Note It seems you are saying there are more than 1 conduit?  It looks like your figure 1 may have more than one vent but are you convinced that there is more than 1 conduit?  If there is evidence you might highlight this more clearly in the text to speak about a ‘conduit structure’ instead.

391 started its activity….created produced

392 to < until

392 started to work…. reached its peak 

394 After this moment, LP activity only occurred infrequently.

404 …anomaly, representing the magma chamber, was located…

405 ‘In the cover’….do you mean ‘in the overlying carapace’

405 ‘was interpreted’  for your audience you are interpreting it at the written moment….. Suggest using present tense  ‘is interpreted’

405-407 This passage is confusing as the phase transition may be interpreted as in the magma (2 phase bubbles and melt) or in the overlying hydrothermal system (2-phase gas and fluid water)….I may be mis-interpreting what you are trying to say so I think your audience will be confused also….. I might interpret what you are saying as the high vp/vs as the partial melt or magma, and the low vp/vs as an overlying magma carapace with fluids from the magma propagating through this conduit root..  If this is your meaning, you could modify the language to suit….Alternatively you may be interpreting the LP as coming from the top of the frothed magma reservoir (within the magma itself) or as magmatic dendrites but it is a bit unclear to me.  Regardless, can you please assess this passage for clarity so your audience will understand what you are trying to say?

Discussion-  It might be worth while to discuss why you think the LP earthquake families show the observed evolution? What is causing illumination of one part of the system and then another?  I suspect that rise of magmatic gasses must overcome a constriction in the conduit to progress and this allows the evolutionary process to proceed.  For example, pressure is overcome at a constriction (LP family) and gas proceeds to a new constriction (new LP family) or if it cannot overcome a constriction it may proceed laterally until a new pathway is established and a new constriction (LP family) is encountered.  As an alternative, my point about a family evolving into a new family might be interpreted as the slow migration of the source along a constricted pathway.  I think this concept is discussed in Park et al. paper discussed above and might be a good discussion point for your paper?

Author Response

Point 1: Citing more papers beneficial for the study.

Response 1: I've found six papers proposed by you very interesting. So much so, that it gave us further directions for a more sophisticated analysis of data and results, which may probably be in the basis of a whole new paper in the future. For the proposed minor revisions in the current article, I've added the following citations in the references section:

  1. Hurst, T.; Jolly, A.D.; Sherburn, S. Precursory characteristics of the seismicity before the 6 August 2012 eruption of Tongariro volcano, North Island, New Zealand. J. Volcanol. Geotherm. Res. 2014, 286, 294–302, doi:10.1016/j.jvolgeores.2014.03.004.
  2. Battaglia, J.; Métaxian, J.-P.; Garaebiti, E. Short term precursors of Strombolian explosions at Yasur volcano (Vanuatu). Geophys. Res. Lett. 2016, 43, 1960–1965, doi:10.1002/2016GL067823.
  1. Petersen, T. Swarms of repeating long-period earthquakes at Shishaldin Volcano, Alaska, 2001–2004. J. Volcanol. Geotherm. Res. 2007, 166, 177–192, doi:10.1016/j.jvolgeores.2007.07.014.
  1. Jolly, A.D.; Neuberg, J.; Jousset, P.; Sherburn, S. A new source process for evolving repetitious earthquakes at Ngauruhoe volcano, New Zealand. J. Volcanol. Geotherm. Res. 2012, 215216, 26–39, doi:10.1016/j.jvolgeores.2011.11.010.
  2. Park, I.; Jolly, A.; Kim, K.Y.; Kennedy, B. Temporal variations of repeating low frequency volcanic earthquakes at Ngauruhoe Volcano, New Zealand. J. Volcanol. Geotherm. Res. 2019, 373, 108–119, doi:10.1016/j.jvolgeores.2019.01.024.

Point 2: Additional discussion about the off-diagonal part of CC matrix

Response 2: The text in section "Cluster analysis" and "Discussion" was reworked. I think the implementation of some of the methods used in suggested papers would require major changes in manuscript and especially figures. Thus, similarily to point 1 - this discussion gives us plenty of ideas for future work on the project, however it cannot be done in five days given for 'minor changes' status of the submission.

Point 3: It might be worth while to discuss why you think the LP earthquake families show the observed evolution? What is causing illumination of one part of the system and then another?

Response 3:

Smooth transitions in the number of detected events between families (Figure 10) and the structure of CC matrix (Figure 7(a)) imply that each dominant family gradually evolves into the next one. This may be interpreted as the slow migration of the source along a constricted pathway of magmatic gasses ascent. Close likelihood of template waveforms for master events of “blue”, “green”, “orange” and “purple” families also support this point, while “red” one represents alternative explanation. The co-existence of two LP families acting in the same period (but not simultaneously) may be interpreted as the balancing stage of the gas ascent process. In such conditions, the pressure gradient allows only a portion of gas bubbles to overcome constriction in the “blue” family origin point, while the rest of the gasses have to proceed laterally (Figure 8(c)) to the “red” family origin point.

Reviewer 2 Report

In line 55 after waveform add “allowing to reconstruct the source geometry (e.g. Battaglia et al., 2003; Rowe et al., 2004; Green and Neuberg, 2006, Gambino, 2006; Gambino et al., 2009)

Battaglia, J., Got J.L., Okubo P. (2003): Location of long-period events below Kilauea Volcano using seismic amplitudes and accurate relative relocation, J. Geophys Res., 108, 2553, doi:101029/2003JB002517.

Rowe, C.A., Thurber C.H., White R.A. (2004): Relocation of volcanic event swarms at Soufriere Hills volcano, Montserrat, 1995-1996, J. Volcaniol. Geotherm. Res., 134, 199-221

Gambino, S. (2006): High precision locations of LP events on Mt. Etna: reconstruction of the fluid-filled volume, Stud. Geoph. Geod., 50, 663-674.

Green, D., Neuberg J. (2006): Waveform classification of volcanic low-frequency earthquake swarms and its implication at Soufrière Hills Volcano, Monserrat., J. Volcanol. Geotherm. Res., 153, 51-63.

Gambino, S., Cammarata, L., Rapisarda, S., (2009). High precision locations of long-period events at La Fossa Crater (Vulcano Island, Italy). Annals of Geophysics, 52, 2, 137-147.

Increase the font size in figures 3,4 and 5.

Author Response

Very valuable references were added and cited in lines 55-56 with the addition of "allowing to reconstruct the source geometry" part at the end of the sentence, which makes it more complete.

The font size was increased as requested in figures 3, 4, and 5, hopefully making it easier for the audience to view presented results.

Please see the attachment with MS Word "Track changes" feature turned on.

Round 2

Reviewer 1 Report

Review of Geosciences 811652

By Abramenkov et al.

The manuscript is much improved from the prior version.  I will note some very minor suggestions for the authors in the manuscript at points of the modified text.

Art

124        ‘The last (post-caldera)…’

127        ‘During the Holocene...’

130        represent < present   suggestion???